# Efficient access to general α-tertiary amines via water-accelerated organocatalytic multicomponent allylation

Prithwish Goswami[1,2], Sung Yeon Cho[1,2], Jin Hyun Park[1,2], Woo Hee Kim[1], Hyun Jin Kim[1], Myoung Hyeon Shin[1] & Han Yong Bae [1✉]

A tetrasubstituted carbon atom connected by three $sp^3$ or $sp^2$-carbons with single nitrogen, i.e., the α-tertiary amine (ATA) functional group, is an essential structure of diverse naturally occurring alkaloids and pharmaceuticals. The synthetic approach toward ATA structures is intricate, therefore, a straightforward catalytic method has remained a substantial challenge. Here we show an efficient water-accelerated organocatalytic method to directly access ATA incorporating homoallylic amine structures by exploiting readily accessible general ketones as useful starting material. The synergistic action of a hydrophobic Brønsted acid in combination with a squaramide hydrogen-bonding donor under aqueous condition enabled the facile formation of the desired moiety. The developed exceptionally mild but powerful system facilitated a broad substrate scope, and enabled efficient multi-gram scalability.

[1] Department of Chemistry, Sungkyunkwan University, Suwon 16419, Republic of Korea. [2] These authors contributed equally: Prithwish Goswami, Sung Yeon Cho, Jin Hyun Park. ✉email: hybae@skku.edu

The α-tertiary amine (ATA) is the key structural motif of a tetrasubstituted carbon atom connected by three $sp^3$ or $sp^2$-carbons with a single nitrogen atom. Substances with the ATA moiety are frequently encountered in numerous biologically active natural products and pharmaceuticals[1,2] (Fig. 1a). Owing to its importance, a few catalytic approaches have been developed to obtain this structure including amination of alkene, C–H alkylation of nitrogen, introduction of amines in tertiary alkyl compounds, and 1,2-addition of carbon nucleophiles to ketimines[3,4]. Among the established advances, an allylative carbon–carbon

bond forming reaction[5,6] between a nucleophilic allylboronate reagent[7] and ketimine is one of the most straightforward processes. Considerable attention has been paid to this approach because of the excellent affinity toward carbonyl-derived ketimines, high reactivity, and relatively low toxicity[8]. Although allylboration processes that provide α-secondary amines through aldimine acceptors are well established, surprisingly few are known to access ATA-incorporating homoallylic amines via ketimine formation processes[5]. One reason could be ascribed to the low reactivity originated from the steric hindrance and the

**Fig. 1 Importance of the α-tertiary amine (ATA) containing molecules and the catalytic allylboration approach toward ATA. a** ATA moiety incorporated pharmaceuticals. **b** Catalytic approaches of ATAs and allylboration toward homoallylic amine. **c** Challenges in the current catalytic allylboration. **d** Hydrophobic amplification under aquacatalytic system. **e** Multi-component organocatalytic allylation (this work). Ac = acetyl; Bn = benzyl; t-Bu = tert-butyl; Bz = benzoyl; cat. = catalyst; HMDS = bis(trimethylsilyl)amide; Me = methyl; Ph = phenyl.

electron-donating nature of connected alkyl/aryl groups. Also, another may be the relatively lower stability of the ATA-incorporating homoallylic amine moiety in the catalytic system favors the retro-transformation to the starting ketimine (or even further to ketone) rather than the desired forward ATA formation process. More notoriously, the rapid tautomerization of enolizable ketimine toward unreactive enamine-type species under the catalytic reaction condition hampers desired constructive pathway (Fig. 1c)[9]. Therefore, an appropriately designed catalytic approach involving reactive components may be necessary for achieving stable products.

To this end, the Shibasaki et al. reported the copper-catalyzed allylboration of N-benzyl ketimine in tetrahydrofuran (THF) and achieved quantitative chemical yields[10,11]. Kobayashi's group introduced zinc-amine as an effective catalyst along with acylhydrazone as a structural equivalent of ketimine in a pentane-THF solvent mixture[12] (Fig. 1b). Regarding the three-component allylboration chemistry, Schaus et al. used wide variety of aldehyde acceptors[13], and Zhang et al. used isatin as ketone acceptor[14] under anhydrous $CH_2Cl_2$ conditions. Despite these promising advances, a suitable protocol involving unexplored ketone substrates (e.g., β-ketoesters, trifluoromethyl ketones, and conjugated ketones) as in situ accessible ketimine precursors and allowing for preparative scalability is still elusive. In addition, a chemical process utilizing water-compatible catalytic (aquacatalytic) approach is unknown (Fig. 1c).

Water plays a crucial role not only as a unique reaction medium but also as a useful reagent in biochemical transformations that support life[15]. For instance, owing to its ubiquity, inexpensiveness, nonflammability, and nontoxicity, water has been regarded as an ideal reaction medium for green chemical processes[16–22]. Recently, in addition to these general advantages, the unique phenomenon based on the use of water as a "reaction enforcer"[23] has attracted tremendous attention within the scientific community. Specifically, biphasic water-rich condition, i.e., the so-called "on-water" systems[24], reactants with rare solubility in water may generate a hydrophobic hydration shell[25]. The desired transformation proceeds with the generation of a form surrounded by hydrogen-bonding networks of water molecules. The confined water-induced cage is subjected to tremendous pressure, allowing the chemical reaction to be more efficient, similar to that induced by the actual high-pressure effect[26,27]. In some cases such as [4 + 2] cycloaddition[24], conjugate addition reaction[28,29], and protonation reaction[27], remarkable increases in reactivity and/or selectivity, namely, "hydrophobic amplifications"[30] was observed. Particularly, metal-mediated (mainly zinc, indium, tin, and others) bimolecular allylation reactions including organoborons, in which two active reaction partners and one (or more) catalyst are involved in the transition state to promote the reaction, have been widely studied[31]. Mechanistically, Pirrung et al. explained that a chemical reaction with a negative volume of activation ($\Delta V^{\ddagger}$, the difference in the volume of the transition state and starting material is <0) is likely to be accelerated through the aqueous medium effects[32]. It is evident that the aspect becomes more complex for multicomponent processes with three or higher number of reaction partners participating in the reaction. Therefore, the rational design of efficient catalysts is a key factor to achieve the desired chemo- and site-selectivities, as well as suitable reactivity. In this context, the Petasis group developed catalyst-free alkenylation using alkenylboronic acids[33], and the Li group reported gold-catalyzed alkynylation via powerful Mannich-type multicomponent couplings[34] with aldehyde acceptors in water. To date, however, water-compatible multicomponent allylboration reactions to afford ATA moieties from ketone acceptor have not been developed (Fig. 1d).

Herein, we report the significant hydrophobic amplification of a catalytic three-component organocatalytic allylation to directly access to the ATA moiety, which exploits abundant ketones as starting materials. The synergistic action of a hydrophobic Brønsted acid in combination with a squaramide hydrogen-bonding donor (HBD) activator catalytically enables the efficient formation of a challenging ATA moiety under aqueous conditions. The developed catalytic system facilitates a broad substrate scope and enabled efficient multi-gram scalability (Fig. 1e).

## Results

**Establishment of aquacatalytic multicomponent allylation reaction**. Our main objective was to develop an allylboration reaction of in situ-formed ketimines from general ketones. Initial studies focused on performing such multicomponent reaction in the presence of challenging and unexplored substrates (Table 1). As a model reaction, we selected ethyl (2,3,4,5-tetrafluorobenzoyl) acetate **1a** as the starting acceptor (β-ketoester is rarely explored, challenging class of ketone in the whole allylation chemistry)[35] due to its excellent analytical profiles ([1]H NMR and thin layer chromatography), benzhydrazide **2a** as the amine source, and allylboronic acid pinacol ester **3a** as the allylating reagent. At the outset, the reaction was conducted in the absence of a catalyst in $H_2O$–NaCl (sat.) (brine), which is a widely employed fundamental medium used in "on-water" chemistry[36]. Due to the biphasic nature, vigorous stirring (rpm > 1000) was helpful for efficient mixing and consistent outcome[26]. Here, the allylation of ketone led only to 22% NMR yield of undesired compound **5** (60 °C, 24 h, entry 1). A BINOL catalyst [(+)-1,1-bi(2-naphthol), (S)-form][37], which is known to be an effective catalyst for the allylboration of ketones, did not work under aqueous conditions (0% of **4a**, 30% of **5**: entry 2). Carboxylic acids such as benzoic acid (BzOH) or erucic acid exhibited almost no effect on the reaction (<5% of **4a**, 24–32% of **5**: entries 3 and 4).

Interestingly, although undesired side product **5** was partially generated (15%), the desired chemoselective three-component allylation to provide ATA **4a** in a meaningful yield, by using BINOL-derived racemic phosphoric acid (±)-PA (46%, entry 5). The use of a stronger Brønsted acid[38,39] than carboxylic acids, i.e., trifluoromethanesulfonimide ($Tf_2NH$), afforded a mixture of **4a** and **5** (63 and 20%, respectively, entry 6). Scandium tris(dodecylsulfate) (ScTS)[40], the well-known water-compatible metallic Lewis acid, did not perform well in the allylation reaction (entry 7). Another strong super-acid trifluoromethanesulfonic acid (TfOH) was also unsuitable catalyst (**4a/5** = 9%/22%, entry 8). The use of perfluorobutanesulfonic acid (PFBS) led to a moderate yield of the product **4a** (43%); however, substantial side reactions occurred (23% of **5**, entry 9). Nafion (the fluorous polymer acid catalyst) and 4-SCA (4-sulfocalix[4]arene), exhibited no chemoselectivity and afforded low conversions (entries 10 and 11). Furthermore, p-toluenesulfonic acid (PTSA), a widely utilized acid[41] gave homoallylic alcohol **5** as a major component (**4a/5** = 3%/25%, entry 12). Here, very interestingly, the same functional moiety with an extended alkyl chain incorporated acid catalyst, 4-dodecylbenzenesulfonic acid (DBSA)[42] exhibited remarkably outstanding catalytic activity and chemoselectivity, and desired allylation product **4a** was obtained in a considerably high yield (**4a/5** = 62%/12%, entry 13). Further experiments proved that the removal of NaCl salt enhanced the chemoselective formation of the desired product with higher yield and selectivity (**4a/5** = 77%/3%, entry 14). Changing the reaction medium to a conventional bulk organic solvent such as toluene facilitated excellent chemoselectivity; however, a low yield was obtained (**4a/5** = 45%/0%, entry 15).

Inspired by the previous discovery that allowed for excellent anion-binding affinity between the squaramide HBD and sulfonate anion[43,44], we additionally employed analogs of diverse (thio)squaramides (SQA-I, II, and III). Squaramide SQA-I[45] did

**Table 1 Establishment of the organocatalytic multicomponent allylation.**

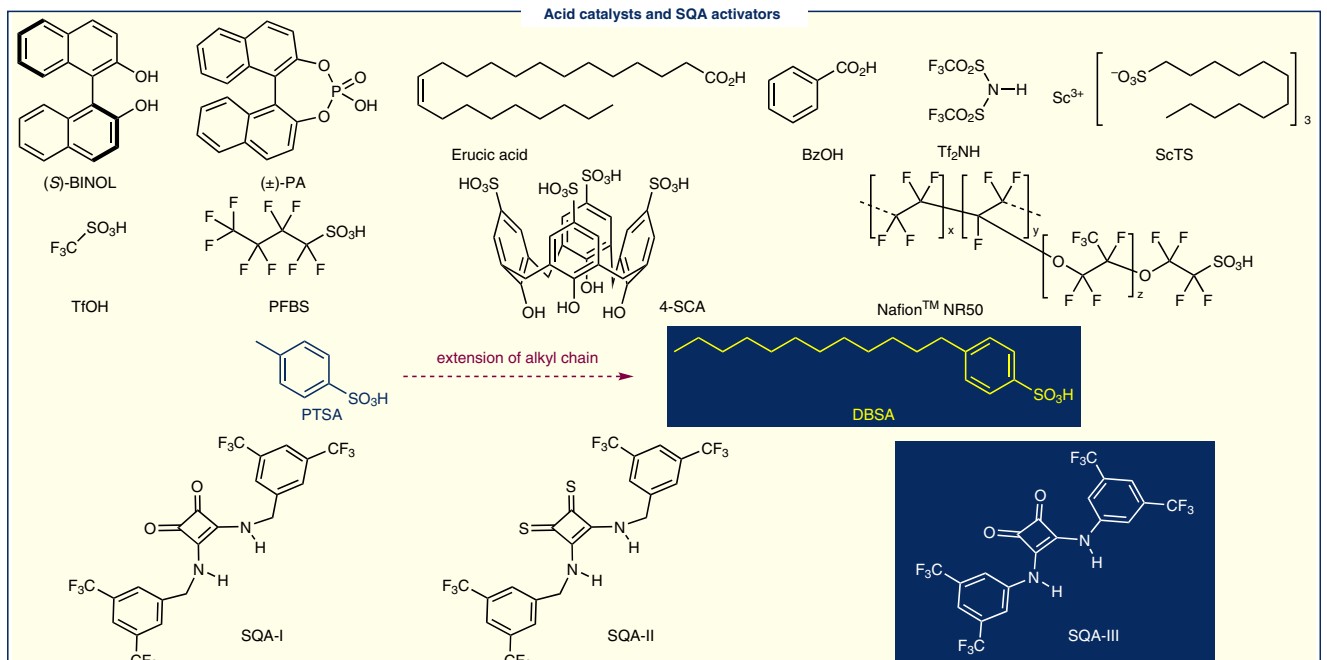

| Entry | Acid catalyst (5 mol%) | Reaction medium | Activator (5 mol%) | Additive (x equiv.) | Yield (%) 4a | Yield (%) 5 |
|---|---|---|---|---|---|---|
| 1 | - | H₂O-NaCl (sat.) | - | - | n.d. | 22 |
| 2 | (S)-BINOL | H₂O-NaCl (sat.) | - | - | n.d. | 30 |
| 3 | BzOH | H₂O-NaCl (sat.) | - | - | <5 | 24 |
| 4 | Erucic acid | H₂O-NaCl (sat.) | - | - | n.d. | 32 |
| 5 | (±)-PA | H₂O-NaCl (sat.) | - | - | 46 | 15 |
| 6 | Tf₂NH | H₂O-NaCl (sat.) | - | - | 63 | 20 |
| 7 | ScTS | H₂O-NaCl (sat.) | - | - | 67 | 24 |
| 8 | TfOH | H₂O-NaCl (sat.) | - | - | 9 | 22 |
| 9 | PFBS | H₂O-NaCl (sat.) | - | - | 43 | 23 |
| 10 | Nafion | H₂O-NaCl (sat.) | - | - | 27 | 25 |
| 11 | 4-SCA | H₂O-NaCl (sat.) | - | - | 14 | 34 |
| 12 | PTSA | H₂O-NaCl (sat.) | - | - | 3 | 25 |
| 13 | DBSA | H₂O-NaCl (sat.) | - | - | 62 | 12 |
| 14 | DBSA | H₂O | - | - | 77 | 3 |
| 15 | DBSA | PhMe | - | - | 45 | n.d. |
| 16 | DBSA | H₂O | - | PhMe (5) | 81 | 2 |
| 17 | DBSA | H₂O | SQA-III | - | 88 | 4 |
| 18 | - | H₂O | SQA-III | PhMe (5) | 1 | 8 |
| 19 | DBSA | H₂O | SQA-I | PhMe (5) | 83 | 2 |
| 20 | DBSA | H₂O | SQA-II | PhMe (5) | 17 | 6 |
| 21 | DBSA | H₂O | SQA-III | PhMe (5) | 96 | 4 |

[a]Conditions: Reactions were performed using ketone **1a** (0.2 mmol), benzhydrazide **2a** (1.2 equiv.), allylboronic acid pinacol ester **3a** (1.5 equiv.), and a set of catalysts (5.0 mol%) in medium (2.0 mL: 10 L/mol (**1a**)) and additive (5.0 equiv.) at 60 °C for 24 h. [b]Yields were determined by ¹H NMR analysis using 1,3,5-trimethoxybenzene as the internal standard. [c]n.d. = not detected. Bz = benzoyl; cat. = catalyst; Et = ethyl.

not show a significant difference in terms of reactivity and chemoselectivity (**4a/5** = 83%/2%, entry 19). A structural analog, thio-squaramide SQA-II[46] significantly decreased the reactivity (**4a/5** = 17%/6%, entry 20). Surprisingly, however, SQA-III[47] proved to be the optimal activator for the reaction outcomes: 96% yield of the desired product **4a** with 4% yield of side product **5** under identical reaction conditions (entry 21). It is worth to note that the catalytic activity was more increased when 5 equivalents of PhMe was used as a hydrophobic additive[48] (entries 14, 16, 17, and 21). The use of the SQA-III without DBSA did not have any effect on the reaction (entry 18).

**Study of the reaction media, amine sources, and allylboronates.** A detailed condition optimization study of the reaction medium was conducted; the results are shown in Fig. 2a. Organic solvents such as toluene (PhMe), dichloromethane ($CH_2Cl_2$), diethyl ether ($Et_2O$), THF, and acetonitrile (MeCN) exhibited low to moderate efficiencies (19–45% of **4a**, entries A–E). The use of bulk toluene with water as an additive (5 equiv.) did not serve to be an effective condition (**4a/5** = 33%/3%, entry F). Environmentally benign green media such as ChCl-urea (1/2 = mol/mol mixture of choline chloride/urea) and ChCl-Gly (1/2 = mol/mol mixture of choline chloride/glycerol) deep eutectic solvents (DESs)[49] were not efficient for this reaction (1 and 33% yields of **4a**, entries G and H, respectively).

We then decided to employ a water-rich reaction medium for this allylation reaction. Designer surfactants[50] SPGS-550-M and TPGS were not effective, as they afforded only 11 and 15% yields of **4a** (entries I and J), respectively. In addition, we found that an anti-hydrophobic agent[51] $LiClO_4$, afforded a lower yield than that of hydrophobic agent, NaCl (46 and 62% yields, entries K and L, respectively). These results support the fact that the reaction may rarely proceed under micelle-promoted aqueous conditions, but it is more likely to be accelerated under biphasic "on-water" conditions. Finally, we found that a small amount of hydrophobic additives, such as 1.0 to 10 equiv. of toluene in the presence of activator SQA-III allowed for the highest reaction outcome in terms of both chemical yield and chemoselectivity (92–96% yield of **4a**, with <4% yield of **5**, entries O–Q).

The selection of a suitable amine source played a crucial role in the success of the devised protocol (Fig. 2b). The summarized results were obtained by reacting ketones (**1a** and **10h**) with a wide variety of amines (**2a**–**2n**). When benzhydrazide (**2a**) was substituted with 4-nitro, 4-methoxy, and 4-methyl substituents (**2b**–**2d**) in the model reaction, lower yields of the corresponding allylation products were obtained (76–87% yields). The reaction mixture of **1a** with *p*-toluenesulfonyl hydrazide (**2e**) underwent decomposition with no detectable product formation. The use of aniline-type compounds such as 2-aminophenol **2f** and *o*-phenylenediamine **2g** afforded only homoallylic alcohol **5** (29% and 9%, respectively). 2-Aminothiophenol **2h** was unexpectedly converted to *N,S*-acetal **9** in 65% yield, with no allylation product formed. Other amines (*t*-butyl or benzyl carbamates, ammonia, hexamethyldisilazane, and Ellman's sulfinamide[52] **2i**–**2m**) were not suitable for the desired three-component allylation. Therefore, a water-compatible, reactive benzhydrazide (**2a**) is highly essential for the success of the desired allylation process.

An identical general reaction trend was observed in the presence of simple ketones. The applicability of an alkyl-alkyl ketone, such as benzylacetone (**10h**), as a representative general acceptor was investigated. Here, hydrazide **2a** also was proved to be an optimal aminating source, providing the desired allylation product **11h** in excellent yield (90% yield). Other amines were not suitable reactions partners for the desired multicomponent allylation reaction and provided a homoallylic alcohol **12**.

Furthermore, the applicability of other structural analogs of allylboronate as donors was verified (Fig. 3). Allylboronic acid pinacol ester (**3a**) proved to be an optimal reagent under the established conditions. Relatively inferior efficiencies were obtained for other allylic analogs (**3b** and **3c**: 87 and 91% yields, respectively). In contrast, potassium allyltrifluoroborate **3d** was not suitable under aquacatalytic conditions (38% of **4a**). Other boronate nucleophiles such as 3-methylbut-2-enylboronic acid pinacol ester and allenylboronic acid pinacol ester were inactive under the established catalytic conditions.

**Generality of aquacatalytic allylation reaction.** A wide variety of carbonyl compounds was investigated under the optimized aqua-catalytic conditions. First, we investigated a series of β-ketoesters (**1a**–**1i**, Fig. 4a). As described herein, fluorinated aromatic sub-stituted compounds (such as **1a** and **1h**) as well as other aryl- and alkyl-substituted β-ketoesters were smoothly converted to the desired homoallylic amine precursors (**4a**–**4i**) in almost quantita-tive yields (up to >99%). Almost no detectable side products were observed. To investigate the relative reactivity regarding the med-ium effect, some selected β-ketoesters were also reacted in toluene as the solvent instead of water (**4a** and **4c**). As a result, significantly lower reactivities were obtained (43–51% yields). In some cases, a lower catalyst loading of 2 mol% and milder reaction temperature (25 °C) were also sufficient for the reaction to undergo completion. Simple alkyl-alkyl ketones (**10a**–**10i**) and alkyl-aryl ketones (**14a**–**14i**) were also smoothly converted to the corresponding desired products (**11a**–**11i** and **15a**–**15i**, respectively) in good to excellent yields (up to 96%, Fig. 4b, c). Similar rate-deceleration effect was observed in toluene (**11d**, **15a**, **15d**–**15i**, 17–69%). Interestingly, in the case of conjugated ketones (**16a**–**16c**), all starting materials were smoothly converted to the desired products in moderate to good yields (**17a**–**17c**, 55–80%). No detectable 1,4-adduct or other side products were observed (Fig. 4d).

The trifluoromethyl ($CF_3$–) group plays a privileged role in medicinal chemistry. Because its substitution into bioactive compounds enhances efficacy by promoting electrostatic interac-tions with targets. Moreover, it improves cellular membrane permeability and enhances the robustness of oxidative metabolism[53,54]. Such trifluoromethyl group incorporated com-pounds are even challenging to prepare and rarely explored in allylation chemistry, owing to their specific nature and reactivity[55]. It was previously reported that starting from an aldehyde, several redox steps including (i) benzoyl hydrazone formation, (ii) C–H trifluoromethylation, and (iii) allylation in the presence of an excess amount of zinc could provide the target homoallylamine derivative[56]. However, under our multicomponent allylation conditions, the desired $CF_3$-substituted products could be obtained up to 89% yield (**19a**–**19d**) via a single aquacatalytic operation, starting from readily available 2,2,2-trifluoroacetophenone analogs (**18a**–**18d**, Fig. 5a). Here, it is noteworthy that hydrazone intermediates were detected and identified as stable form. Isolated $CF_3$-hydrazones are further active in the developed standard allylation reaction under the identical condition, and afforded products **19** without any problem. In addition, alkyl- and aryl-aldehyde substrates (**20a**–**20d**) were also converted to the desired α-secondary amine incorporated allylation products (**21a**–**21d**, Fig. 5b). The late-stage modification[57] of complex biologically active ketones was possible and smoothly afforded the correspond-ing allylation products (Fig. 5c). Drug derivatives such as sodium salt of warfarin and ketoprofen methyl ester (**22a** and **22b**, respectively) were transformed to the desired products in 40% (**23a**) and 45% (**23b**) yields, respectively. Moreover, a fragrance molecule hedione® (**22c**) was successfully converted to the desired allylation product (**23c**) in 73% yield.

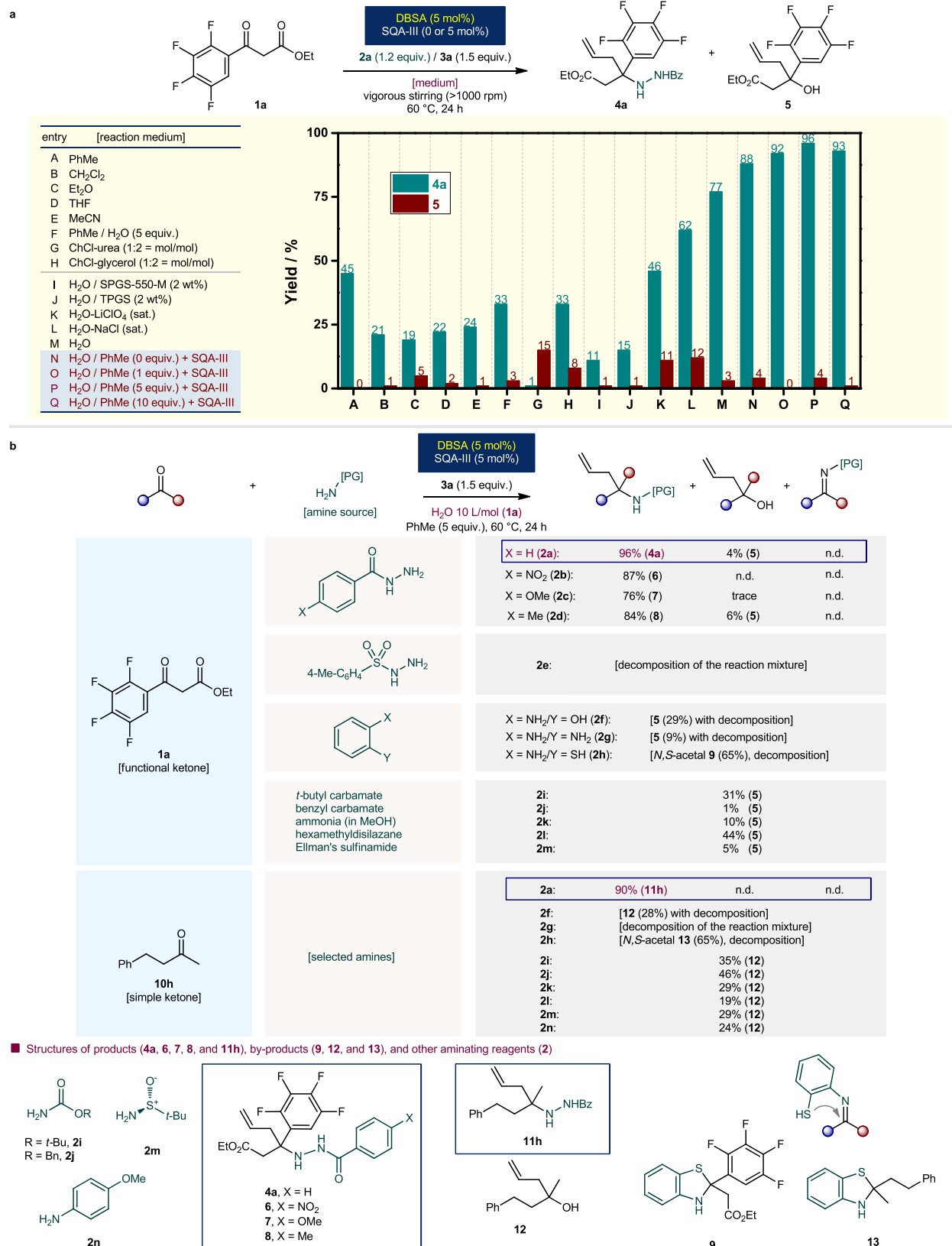

**Fig. 2 Detailed study on the reaction media and amine sources. a** Effect of reaction medium. **b** Detailed study on the amine source. [a]Conditions: Reactions were performed with ketone (0.2 mmol), amine source (1.2 equiv.), allylboronic acid pinacol ester **3a** (1.5 equiv.) and set of catalysts (5.0 mol%) with medium (2.0 mL: 10 L/mol (**1a** or **10h**)) at 60 °C for 24 h. [b]Yields were determined by [1]H NMR analysis using 1,3,5-trimethoxybenzene as internal standard. [c]n.d. = not detected. Bn = benzyl; *t*-Bu = *tert*-butyl; Bz = benzoyl; Et = ethyl; Me = methyl; Ph = phenyl.

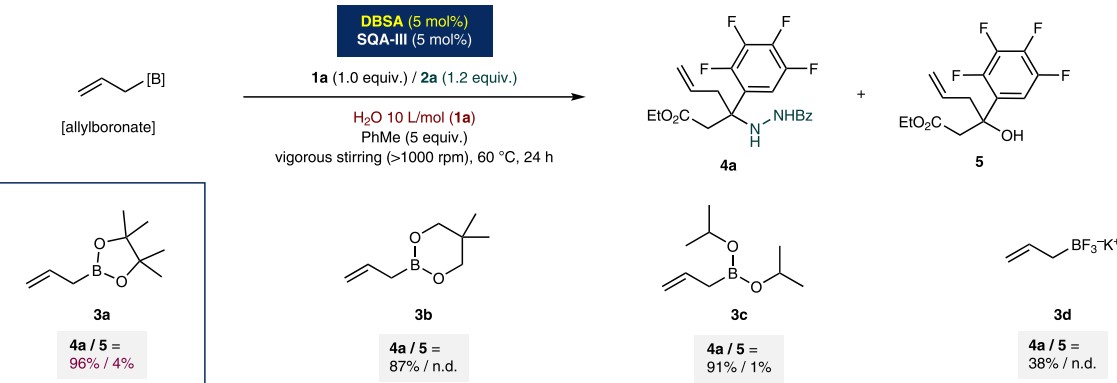

**Fig. 3 Detailed study on the allylboronates.** [a]Conditions: Reactions were performed with ketone **1a** (0.2 mmol), benzhydrazide **2a** (1.2 equiv.), allylboronate (1.5 equiv.) and set of catalysts (5.0 mol%) with water (2.0 mL: 10 L/mol (**1a**)) and PhMe (5.0 equiv.) at 60 °C for 24 h. [b]Yields were determined by [1]H NMR analysis using 1,3,5-trimethoxybenzene as internal standard. [c]n.d. = not detected. Bz = benzoyl; Et = ethyl.

**Synthetic applications of catalytic allylation**. To verify the practicability of the aquacatalytic allylation, preparative-scale syntheses with low catalyst loading were performed (Fig. 6a). For the five selected examples, each ketone **1a** (3.79 mmol), **1b** (7.68 mmol), **10c** (9.98 mmol), **10a** (13.9 mmol), and **14d** (7.45 mmol) was tested at a 1.00 g scale in the presence of the lowest catalyst loading of 0.5 mol%. By conducting the experiments under vigorous stirring (>1000 rpm) for 48 h at 25 or 60 °C, all corresponding allylation products were quantitatively obtained (**4a** = 91% yield/ 1.45 g; **4b** = 99% yield/ 2.20 g; **11c** = 94% yield/ 2.45 g; **11a** = 96% yield/ 3.09 g; **15d** = 90% yield/ 1.97 g) with nearly perfect chemoselectivities, and no isolable side product was identified. In all cases, the reaction proceeded in an organic-aqueous biphasic reaction system (Fig. 6b). The synthetic utility of the obtained homoallylic amine product **15a** was highlighted in Fig. 6c. The treatment of HCl (aq.) provided the salt of de-benzoylated product **24** in low yield (10% NMR yield). However, SmI₂ very successfully provided N–N cleaved free-homoallylic amine **25** with quantitative yield (>99% NMR yield). For further transformation to the useful functionalized molecules, we synthesized Boc-protected amine **27** within simple two steps (Boc protection to compound **26**, then N–N cleavage) from **15a**. The acidic deprotection of Boc-group also can simply afford **25** in 97% yield. The conventional oxidative cleavage of compound **27** could be converted to the carboxylic acid **28**, then further methylation gives the ester **29** in 64% yield (2 steps). Reaction with NBS mediated bromo-cyclization gives the compound **30** in 70% yield. Finally, the N-Boc homoallylic amine **27** was even metathesis-active that conjugated ester **31** was successfully obtained with clean reaction profile (97% yield).

**Investigation of hydrophobic amplification and reaction mechanism**. Experimental, analytical, and computational studies were implemented to elucidate the mechanism of the developed reaction (Fig. 7). Initially, experiments to compare reactivity were conducted using acid catalysts PTSA and DBSA. Optimized (i) bulk water, and (ii) homogeneous bulk toluene conditions were used for each acid catalyst. In bulk toluene, both catalysts showed similar trends with moderate reaction profiles within 24 h (at 24 h, DBSA = 60% yield; PTSA = 41% yield). In sharp contrast, the utilization of "on-water" reaction conditions led to significant differences. In the case of PTSA, the expected allylation reaction did not proceed at all, and SQA-III could not function as an activator (<1% yield). However, in the presence of DBSA, the initial reaction rate was remarkably rapid, and a conversion of 96% was reached within 18 h. To compare the

relative hydrophobicity of the catalysts, the Log $P$ value for each catalyst was computed [$P = n$-octanol/water partition coefficient, using Spartan '14 for Windows (energy/DF/EDF2/6-31G*/ vacuum)]. DBSA and PTSA exhibited a Log $P$ value of +4.78 and +0.19, respectively. These results suggested that the more hydrophobic DBSA preferentially involved in the "confined" organic cage to actively engage in the catalytic transformation, whereas the relatively less hydrophobic PTSA was ineffective for the aquacatalysis (Fig. 7a).

The surprising hydrophobic amplification effect observed under the biphasic "on-water" conditions could be ascribed to the enhancement of the catalytic activity due to strong anion binding between dodecylbenzenesulfonate (the sulfonate anion of DBSA) and squaramide SQA-III. Theoretical and analytical studies on anion binding between squaramide HBD and triflate, i.e., [SQA]:[sulfonate anion] = 1:1 complex, have been extensively conducted by Jacobsen et al.[43,44] Inspired by the strong HBD affinity toward sulfonates, the ground state energy of the corresponding structure was computed. DFT-based geometry optimization of dodecylbenzenesulfonate and SQA-III (B3LYP, 6-31G*, toluene) resulted in a complex, in which the two N–H hydrogen atoms of SQA-III are bound to an oxygen atom of the sulfonate by hydrogen-bonding interaction with additional weak π–π interactions[58]. This complex was found to be more stable by 2.15 kcal/mol than the corresponding complex with $p$-toluenesulfonate, in which a more active hydrophobic catalyst complex was assembled by using DBSA than that by using PTSA (see Supplementary Table 4, Fig. 7b). Furthermore, analytical experiments were helpful to elucidate the strong interaction between DBSA and SQA-III. [1]H NMR analysis was conducted after mixing these compounds at a molar ratio of [DBSA]:[SQA-III] = 1:1 (25 °C, DMSO-$d6$). Thus, the down field shift of the N–H protons of SQA-III (from $\delta$ = 10.64 to $\delta$ = 10.67, 2H) along with the disappearance of the acid peak of DBSA ($\delta$ = 10.36, 1H) were observed. These results supported the occurrence of a strong interaction between DBSA and SQA-III. (Fig. 7c).

The study on the reactivity was further investigated by employing bulk deuterated water (D₂O) as the reaction medium instead of H₂O under identical standard reaction conditions (Fig. 7d). As a result, a marked reaction medium isotope effect was observed. In general, it is known that for reactions carried out using a "on-D₂O (heavy water)" system, the reactivity is noticeably reduced compared to that in H₂O. This phenomenon was explained probably because of the fact that the viscosity of D₂O is higher than that of H₂O[24]. Interestingly, the deuterated product at **4a**-(*d*) was obtained in a significantly lower yield (75%) with >99% deuteration at the α-carbon position. This result

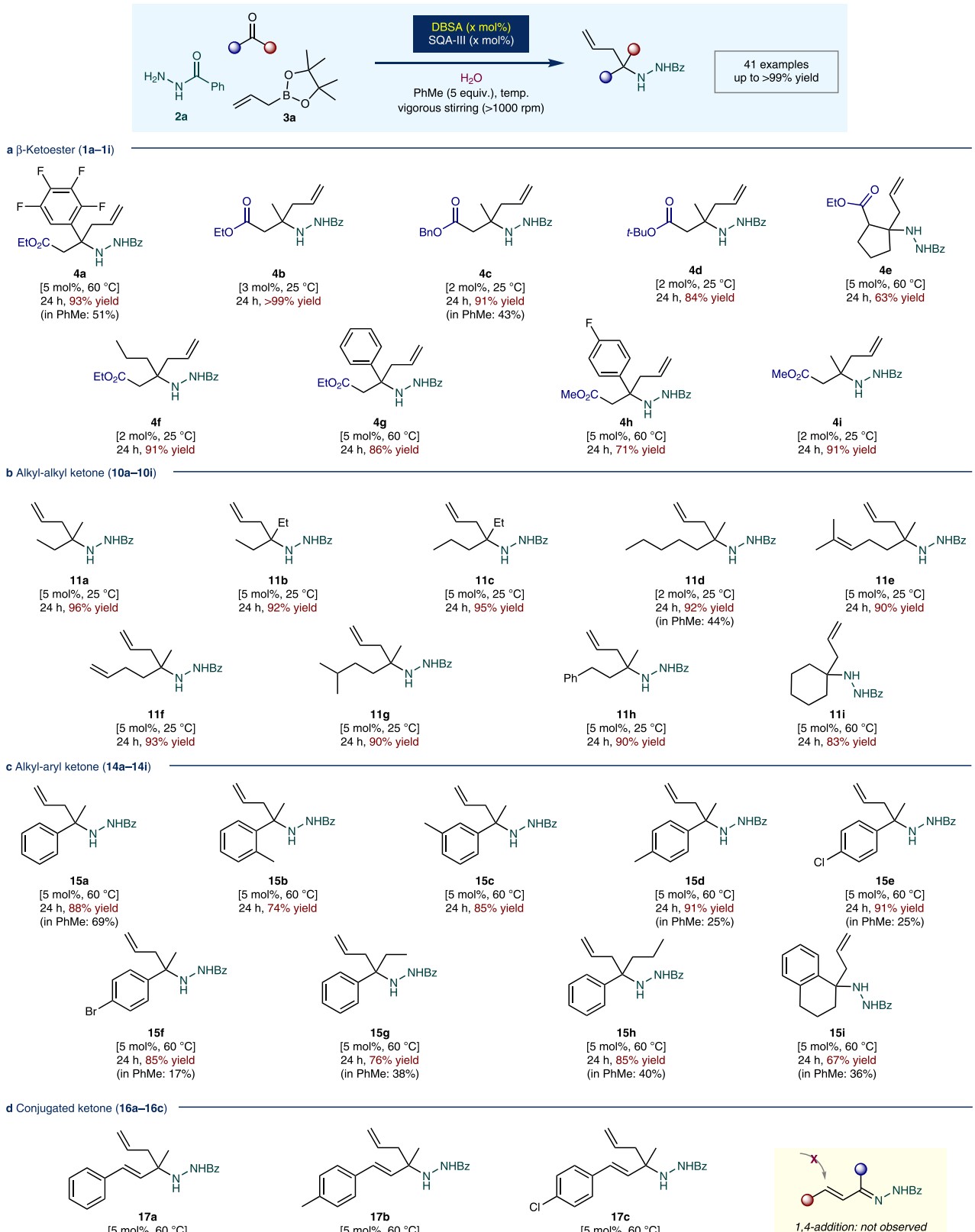

**Fig. 4 Substrate scope of allylation reaction (continued on Fig. 5). a** β-Ketoester. **b** Alkyl-alkyl ketone. **c** Alkyl-aryl ketone. **d** Conjugated ketone.
[a]Conditions: Reactions were performed with ketone (0.2 mmol), benzhydrazide **2a** (1.2 equiv.), allylboronic acid pinacol ester **3a** (1.5 equiv.), and a set of catalysts (2.0–5.0 mol%) with water (2.0 mL: 10 L/mol (ketone)) and PhMe (5.0 equiv.) at 25–60 °C for 24 h. [b]Yields were determined after chromatographic purification. Bn = benzyl; *t*-Bu = *tert*-butyl; Bz = benzoyl; Et = ethyl; Me = methyl; Ph = phenyl.

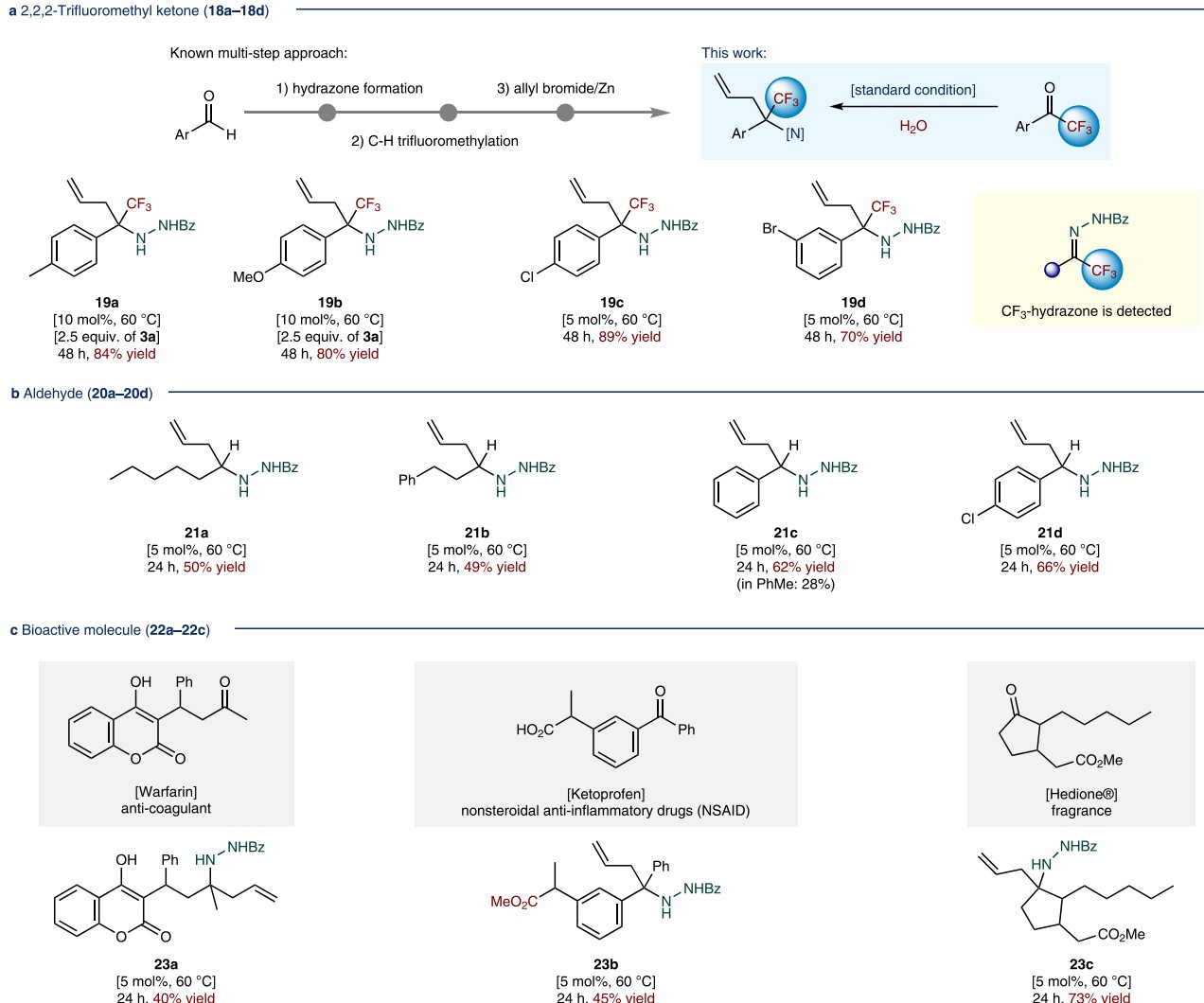

**Fig. 5 Substrate scope of allylation reaction (continued from Fig. 4). a** 2,2,2-Trifluoromethyl ketone. **b** Aldehyde. **c** Bioactive molecule. [a]Conditions: Reactions were performed with ketone or aldehyde (0.2 mmol), benzhydrazide **2a** (1.2 equiv.), allylboronic acid pinacol ester **3a** (1.5–2.5 equiv.), and a set of catalysts (5.0–10 mol%) with water (2.0 mL: 10 L/mol (ketone or aldehyde)) and PhMe (5.0 equiv.) at 60 °C for 24–48 h. [b]Yields were determined after chromatographic purification. Ar = aryl group; Bz = benzoyl; Me = methyl; Ph = phenyl.

supports that reaction proceeded via tautomerized enamine-type intermediate I from in situ generated hydrazone. The side product **5** was not detectable. When ketone **1a** or product **4a** was subjected in D₂O in the absence of benzhydrazide **2a** and allylboronate **3a**, no deuterium exchange occurred.

A plausible reaction mechanism and preliminary working principle of the hydrophobic amplification can be explained as follows. A reactive ketone-derived benzoyl hydrazone is generated in situ by the catalytic action of [DBSA · SQA-III]-based Brønsted acid complex. In principle, it is highly challenging to promote the reaction under conventional organic solution conditions due to the imine-enamine type tautomerization. However, exploiting the "on-water" hydrophobic amplification effect induced by the assembled acid catalysis successfully overcomes the subsequent turnover-limiting step. The allylative carbon–carbon bond formation proceeds through a closed, cyclic Zimmerman–Traxler transition state structure[5,6,59]. At this point, the electrophilicity of the benzoyl hydrazone is increased through Brønsted acid activation (LUMO-lowering), while the nucleophilicity of the allylboronate is enhanced because of the Lewis basic activation of the essential benzoyl group to the Lewis acidic boron center

(HOMO-raising). This synergistic catalysis[60] occurs inside of a hydrophobic hydration shell, namely, a confined water cage, whose formation is induced by the surrounding bulk water. And perhaps a pseudo-high-pressure effect[26,27] is applied by densely formed hydrogen-bonding networks (Fig. 7e).

## Discussion

In summary, we developed a broadly applicable, water-accelerated catalytic multicomponent allylboration reaction to access the ATA structure using in situ generated benzoyl hydrazones. A wide variety of functionalized and simple ketones were employed for in situ hydrazone formation and converted to the corresponding homoallylic amine derivatives with up to >99% yield and very high chemo-/site-selectivities. The synergistic action of a hydrophobic acid catalyst with a squaramide activator successfully enabled this challenging transformation to occur in a preparative-scale under water-rich biphasic conditions. Mechanistic considerations support the suitable hydrophobicity of the employed acid catalyst, which plays a crucial role in achieving such successful transformation. Although preliminary attempt was applied on the asymmetric

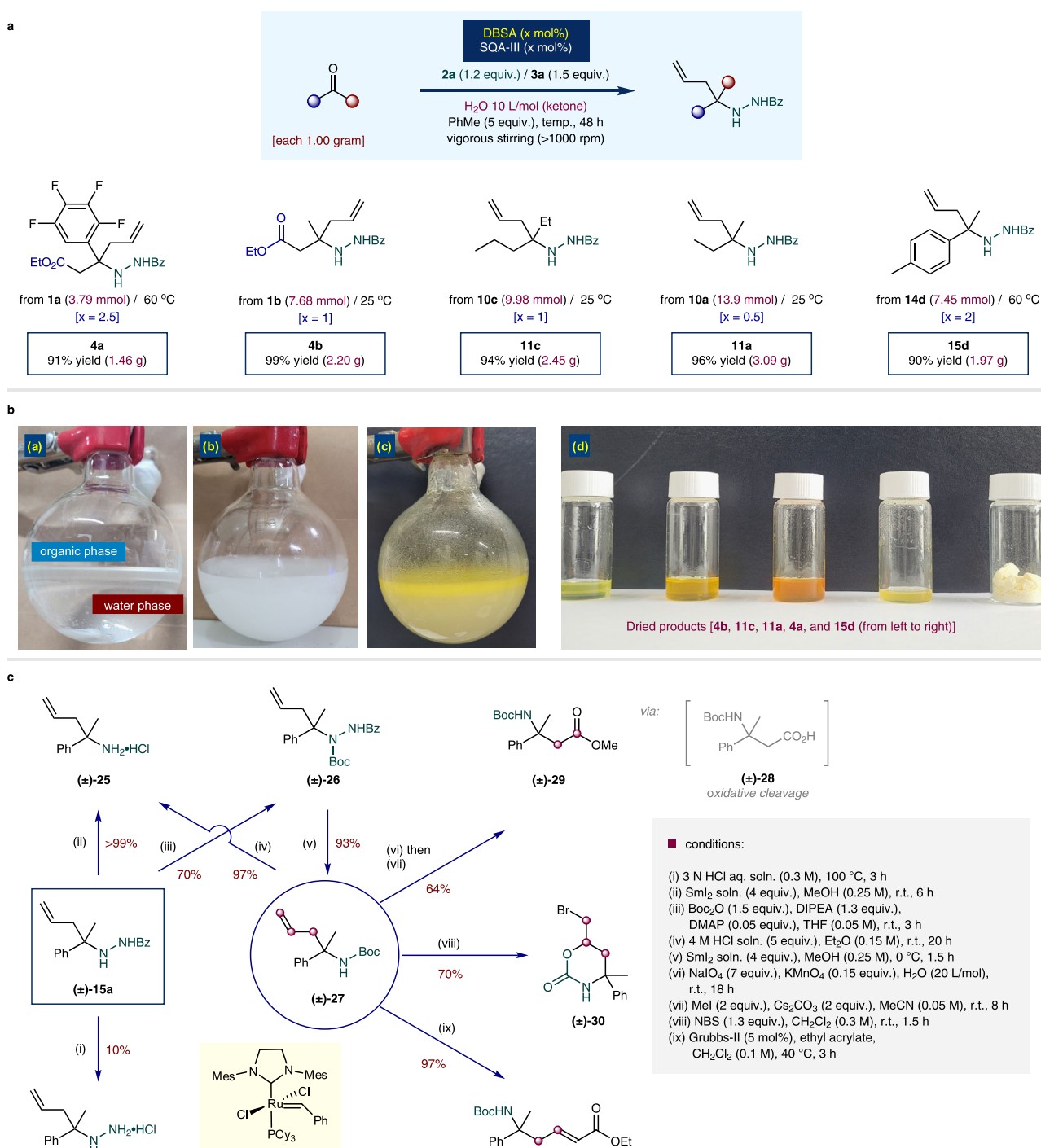

**Fig. 6 Synthetic applications of catalytic allylation. a** Gram-scale synthesis. **b** Pictures of biphasic mixtures and purified products. **c** Synthetic utilities of allylation products. Boc = *tert*-butyloxycarbonyl; Bz = benzoyl; Cy = cyclohexyl; Et = ethyl; Me = methyl; Mes = mesityl; Ph = phenyl.

version by employing chiral anion-binding Jacobsen-type urea[61], however, still a meaningful enantioselectivity was not obtained. In-depth studies to understand more detailed water-acceleration mechanisms and further efforts toward asymmetric processes are ongoing in our laboratory.

## Methods

**General procedures for the allylation reactions**. In a flame-dried capped vial, equipped with multiple magnetic stirring bar and filled with Ar gas, **1a** (0.2 mmol), benzhydrazide (**2a**, 1.2 equiv.), SQA-III (5.0 mol%), and DBSA (5.0 mol%) were added. Subsequently, allyl-Bpin (**3a**, 1.5 equiv.), PhMe (5 equiv.), and H₂O

(deionized, 10 L/mol) were added to the reaction mixture then sealed to stir vigorously (rpm > 1000) at 60 °C for 24 h. The resulting mixture was diluted and extracted with EtOAc/brine, and the combined organic layer was dried over anhydrous Na₂SO₄, and filtered. The filtrate was concentrated in vacuo, and the residue was purified by column chromatography on silica gel to afford corresponding allylation-reaction product (**4a**, 93% yield).

**Typical scale-up experiment (1.0 g scale)**. In a flame-dried capped round-bottom flask, equipped with a magnetic stirring bar and filled with Ar gas, **1a** (1.0 g), benzhydrazide (**2a**, 1.2 equiv.), SQA-III (2.5 mol%), and DBSA (2.5 mol%) were added. Subsequently, allyl-Bpin (**3a**, 1.5 equiv.), PhMe (5 equiv.), and H₂O (deionized, 10 L/mol) were added to the reaction mixture then sealed to stir

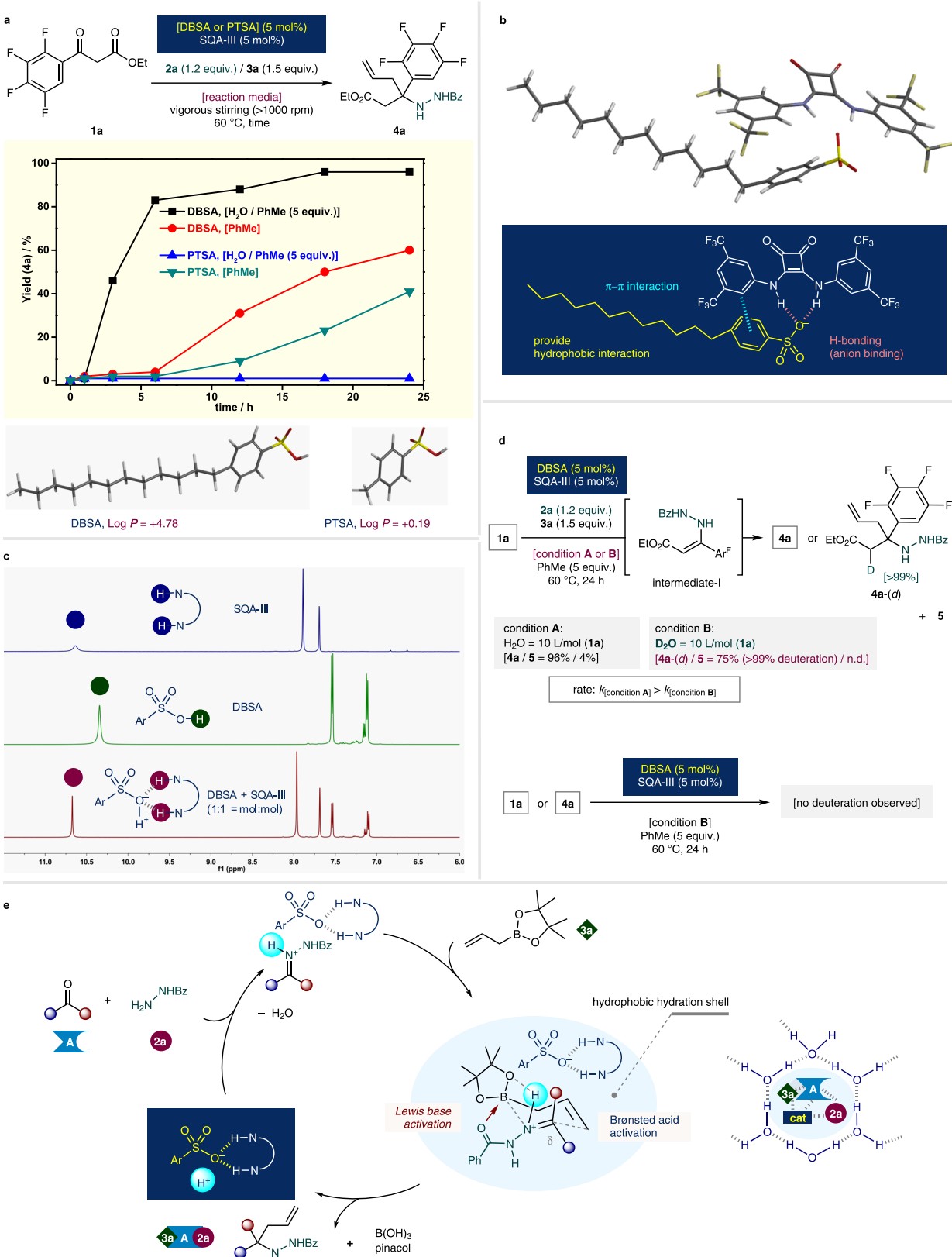

**Fig. 7 Mechanistic investigation. a** Reaction progress and calculated Log $P$ values of the acid catalysts. **b** Complex of SQA-III and sulfonate anion of DBSA (DFT, B3LYP, equilibrium geometry of the ground state in PhMe, 6-31G*). **c** $^1$H NMR spectra of the complexation of DBSA and SQA-III. **d** Reaction media isotope effect. **e** Putative catalytic cycle. **A** = carbonyl compound; Ar = 4-dodecylphenyl; Ar$^F$ = 2,3,4,5-tetrafluorophenyl; Bz = benzoyl; **cat** = DBSA and SQA-III; Et = ethyl.

vigorously (rpm > 1000) at 60 °C for 48 h. The resulting mixture was diluted and extracted with EtOAc/brine, and the combined organic layer was dried over anhydrous $Na_2SO_4$, and filtered. The filtrate was concentrated in vacuo, and the residue was purified by column chromatography on silica gel to afford the corresponding allylation-reaction product (**4a**, 1.46 g, 91% yield).

## Data availability

The authors declare that all relevant data supporting the findings of this study are available within the article and Supplementary Information files, and also are available from the corresponding author upon request.

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

## Acknowledgements

Supports from the Ministry of Science, ICT and Future Planning of Korea (2020R1C1C1006440, 2020R1A4A1018019, and 2021R1A2C2093597), the Institute of Civil-Military Technology Cooperation Center funded by the Defense Acquisition Program Administration and Ministry of Trade, Industry and Energy, and of Korean government (20-CM-BR-05), and the Sungkyunkwan University and the BK21 FOUR (Graduate School Innovation) funded by the Ministry of Education (MOE, Korea) and National Research Foundation of Korea (NRF) are acknowledged. P.G. acknowledges Korea Research Fellowship (2019H1D3A1A01102782). This paper is dedicated to Prof. Benjamin List.

## Author contributions

P.G., S.Y.C., J.H.P., and H.Y.B. developed the reaction, investigated the substrate scope, derivatizations of the allylation products, and implemented analytical studies. W.H.K., H.J.K., and M.H.S. supported experiments and analyses. J.H.P. and H.Y.B. implemented computational studies. H.Y.B., P.G., S.Y.C., and J.H.P. wrote the manuscript. H.Y.B. designed and oversaw the project.

## Competing interests

The authors declare no competing interests.
