## [Peer Review File · Nature Communications]

REVIEWER COMMENTS

Reviewer #1 (Remarks to the Author):

Han Yong Bae et al developed an efficient water-accelerated organocatalytic method to directly access ATA incorporating homoallylic amine structures by exploiting readily accessible general ketones as useful starting material. The unprecedented synergistic action of a hydrophobic Brønsted acid in combination with a squaramide hydrogen-bonding donor under aqueous condition enabled the facile formation of the desired moiety. The developed exceptionally mild but powerful system facilitated a broad substrate scope, and enabled efficient multi-gram scalability. In 2006, Shibasaki reported the Cu-catalyzed enantioselective allylation of ketoimines [J. Am. Chem. Soc. 2006, 128, 7687–7691]. In addition, Schaus et al. reported an asymmetric Petasis Borono-Mannich allylations (three-component allylboration) catalyzed by chiral biphenols [Angew. Chem. Int. Ed. 2017, 56 (6), 1544–1548]. Publication of this manuscript is not recommended in Nature Communications owing to limitation of a symmetric multicomponent allylation, not an asymmetric one.

Reviewer #2 (Remarks to the Author):

In the manuscript by H. Y. Bae et al., an efficient one-pot catalytic methodology to access challenging homoallylic amine structures bearing α -quaternary stereogenic center from readily accessible ketones, hydrazides and allylboronates as starting substrates is reported. The best chemical selectivity and highest yields of allylation products were attained in aqueous micro-emulsions in the presence of hydrophobic Brønsted acid / squaramide catalytic system. It is the first example of the significant hydrophobic amplification of a catalytic three-component organocatalytic allylation to directly access the alpha-tertiary amine moiety using abundant ketones as starting materials. Importantly, the developed procedure is scalable and applicable to a wide range of substrates which makes the methodology sound and reproducible. Experimental results and theoretical calculations support the proposed reaction mechanism.

Therefore, I recommend publication of this manuscript in NC after a few minor corrections listed below are made.

- The authors marked differently two bonds at quaternary carbon atom of allylic products as bold or hashed bonds. Why? All products are racemic and formation of diastereomers is not possible for most of them. To avoid confusions, I recommend the authors to substitute the bold and hashed bonds for regular solid bonds throughout the paper.
- According to general procedures presented in supporting information, the two-phase allylation reactions were carried out under vigorous stirring (rpm > 1000). Is it critical? This point should be somehow discussed in the manuscript.
- Products 26 and 27 may be of insufficient purity (see ¹H NMR spectra in SI). It is advisable to re-purify the compounds to remove side signals.

Sergei Zlotin

Reviewer #3 (Remarks to the Author):

The manuscript by Bae et al. reported a three component reaction for the construction of alpha-tertiary amines, in which acceleration effect of water was found. This is a novel methodology for the synthesis of alpha-tertiary amines from ketone, allylboronate, and hydrazide, albeit such structures (as well as the asymmetric version) were accessible by metal-catalyzed allylboration of ketimine according to previous reports. The broad scope of the ketones makes the methodology useful for synthesis, and the mechanistic information and discussion about the

water effect and hydrophobic activation are clear and interesting. An asymmetric version of the reaction should be expected for a paper in Nat. Commun. but this was not developed in the current work. Overall, the work was well-conducted, and I support the publication of the work in Nat. Commun. after the following concerns are addressed.

(1) The scope of the allylboronate was not investigated to evaluate the substituent effects of groups on the alkenyl moiety. More allylboronates are required.

(2) It's more desirable if general amines could be used, but it was found aminating sources other than hydrazide are not reactive for this transformation. The authors are suggested to give some comments about why the hydrazide is essential in page 7 about the selection of amine source.

(3) According to Figure 5B, the alkyl chain on DBSA is not involved in complexation with SQA-III, why this complex was more stable by 2.15 kcal/mol than the corresponding complex with p-toluenesulfonate?

(4) The abstract needs revision. Too much background information is given in this section.

Reviewer #1 (Remarks to the Author):

Han Yong Bae et al developed an efficient water-accelerated organocatalytic method to directly access ATA incorporating homoallylic amine structures by exploiting readily accessible general ketones as useful starting material. The unprecedented synergistic action of a hydrophobic Brønsted acid in combination with a squaramide hydrogen-bonding donor under aqueous condition enabled the facile formation of the desired moiety. The developed exceptionally mild but powerful system facilitated a broad substrate scope, and enabled efficient multi-gram scalability.

We appreciate this assessment. Our method has high efficiency, general ketone scope, and can be operated under a very mild condition in gram-scale. We believe that even non-expert in synthesis may simply conduct economical three-component organocatalytic reaction to prepare challenging ATA incorporating homoallylic amine according to our protocol.

In 2006, Shibasaki reported the Cu-catalyzed enantioselective allylation of ketimines [J. Am. Chem. Soc. 2006, 128, 7687–7691].

We clearly showed in Fig. 1b, and mentioned the related reference in the main text: Shibasaki and co-workers used organic solvent (THF) and employed pre-formed ketimine with copper/lanthanum metals as catalysts. Also, only simple (aryl-alkyl and alkyl-alkyl) ketimines were working in their study. In our work, however, we improved from previous works and showed a new water-accelerated, metal-free, and synergistic organocatalysis that enabled wider scope and one-pot multi-component reaction. We believe our work is apparently different. More importantly, as we emphasized in Fig. 1c and Fig. 3a, utilizing β -keto ester as starting material in the ketimine allylation is entirely unknown by far.

In addition, Schaus et al. reported an asymmetric Petasis Borono-Mannich allylations (three-component allylboration) catalyzed by chiral biphenols [Angew. Chem. Int. Ed. 2017, 56, (6), 1544–1548].

Although the three-component Petasis borono-Mannich allylation reactions in anhydrous CH_2Cl_2 were reported in the mentioned paper, only aldehydes worked as substrate, and ketone has never been attempted. All the obtained products were α -secondary amines, not α -tertiary amines (ATAs). In addition, as we clearly showed in the experiment using chiral biphenol analog ((*S*)-BINOL), only undesired homoallylic alcohol **5** was synthesized (Entry 2, Table 1). Our protocol is applicable to general ketones, and also covers aldehyde substrates (Fig. 3f). Our work is different from the mentioned reference.

Publication of this manuscript is not recommended in Nature Communications owing to limitation of a symmetric multicomponent allylation, not an asymmetric one.

A non-asymmetric version of such water-accelerated three-component allylation to access challenging ATA is even underdeveloped. Furthermore, exploiting β -keto ester for converting to ATA structure via one-pot catalysis is entirely unknown.

Reviewer #2 (Remarks to the Author):

In the manuscript by H. Y. Bae et al., an efficient one-pot catalytic methodology to access challenging homoallylic amine structures bearing α -quaternary stereogenic center from readily accessible ketones, hydrazides and allylboronates as starting substrates is reported.

We appreciate this assessment.

The best chemical selectivity and highest yields of allylation products were attained in aqueous micro-emulsions in the presence of hydrophobic Brønsted acid / squaramide catalytic system. It is the first example of the significant hydrophobic amplification of a catalytic three-component organocatalytic allylation to directly access the alpha-tertiary amine moiety using abundant ketones as starting materials. Importantly, the developed procedure is scalable and applicable to a wide range of substrates which makes the methodology sound and reproducible. Experimental results and theoretical calculations support the proposed reaction mechanism.

Therefore, I recommend publication of this manuscript in NC after a few minor corrections listed below are made.

We are pleased concerning the positive assessment.

- The authors marked differently two bonds at quaternary carbon atom of allylic products as bold or hashed bonds. Why? All products are racemic and formation of diastereomers is not possible for most of them. To avoid confusions, recommend the authors to substitute the bold and hashed bonds for regular solid bonds throughout the paper.

We are grateful for this valuable comment. To reflect this suggestion, we changed the whole drawing of the chemical structures that appeared in the manuscript and SI. In addition, we revised every detailed point according to *Nature Communications'* formatting guidelines.

- According to general procedures presented in supporting information, the two-phase allylation reactions were carried out under vigorous stirring (rpm > 1000). Is it critical? This point should be somehow discussed in the manuscript.

Vigorous stirring is helpful for the rapid mixing and consistent outcome in the biphasic reaction in general. Particularly, the systematic stirring speed obviously affected the stereoselectivity according to our previous work (Ref. 26: *Nat. Commun.* **10**, 851 (2019)). Fortunately, in this study, we rarely observed a significant effect of the stirring speed on reactivity and chemoselectivity. To reflect this suggestion, we added a sentence in the main text as follows.

“Due to the biphasic nature, vigorous stirring (rpm >1000) was helpful for efficient mixing and

consistent outcome.²⁶

- Products 26 and 27 may be of insufficient purity (see ¹H NMR spectra in SI). It is advisable to re-purify the compounds to remove side signals.

Thank you for this valuable comment. Product **26** is an inseparable mixture of rotamers (ratio = 3.4:1, note: *ratio was corrected in the current SI*) with unstable properties that further analytical characterization (mp and HR-MS) was problematic. Fortunately, this compound was rapidly used in the next step and smoothly converted to compound **27** (no possible rotamer), a stable known compound. Though we tried further experiments to obtain higher purity, the current spectra were the best forms even containing trace amounts of unavoidable NMR solvent impurities. For clarity, we revised the corresponding parts as follows in the SI.

26, new compound, white solid, 803.8 mg, 70% yield (mixture of rotamers).

$R_f = 0.22$ (EtOAc:hexanes = 1:4 v/v).

¹H NMR (500 MHz, CDCl₃, Me₄Si, major and minor isomers): δ 8.32 – 7.88 (m, 1H), 7.86 – 7.76 (m, 2H), 7.75 – 7.65 (m, 2H), 7.57 – 7.47 (m, 1H), 7.46 – 7.37 (m, 2H), 7.33 (t, $J = 7.6$ Hz, 2H), 7.22 (t, $J = 7.2$ Hz, 1H), 5.85 (td, $J = 16.3, 9.1$ Hz, 0.78H), 5.45 (td, $J = 16.8, 8.0$ Hz, 0.23H), 5.31 – 5.06 (m, 1.64H), 5.00 – 4.82 (m, 0.48H), 3.29 – 2.68 (m, 2H), 1.69 (d, $J = 10.9$ Hz, 3H), 1.16 (s, 9H).

¹³C NMR (126 MHz, CDCl₃, Me₄Si, major and minor isomers): δ 168.45, 168.19, 148.42, 146.51, 134.20, 134.11, 132.65, 132.07, 131.99, 128.70, 128.60, 128.06, 127.96, 127.40, 127.31, 126.28, 126.16, 125.91, 125.11, 119.76, 118.11, 81.73, 81.60, 66.57, 66.52, 45.91, 27.93, 27.89, 26.78, 22.96.

§ Note: Inseparable mixture of rotamers (major and minor). Based on the ¹H NMR integration ratio at (δ 5.85, 5.45, 5.31 – 5.06, 5.00 – 4.82), given in non-integer (decimal) values, rotameric ratio was determined as 3.4:1. Due to the unstable properties (*e.g.*, hygroscopicity, decomposition), this compound was immediately used to the next step without further analysis (*e.g.*, mp, HR-MS).

Supplementary Figure 45. ¹H and ¹³C NMR spectra (26)

Reviewer #3 (Remarks to the Author):

The manuscript by Bae et al. reported a three component reaction for the construction of alpha-tertiary amines, in which acceleration effect of water was found. This is a novel methodology for the synthesis of alpha-tertiary amines from ketone, allylboronate, and hydrazide, albeit such structures (as well as the asymmetric version) were accessible by metal-catalyzed allylboration of ketimine according to previous reports. The broad scope of the ketones makes the methodology useful for synthesis, and the mechanistic information and discussion about the water effect and hydrophobic activation are clear and interesting.

We appreciate this assessment.

An asymmetric version of the reaction should be expected for a paper in Nat. Commun. But this was not developed in the current work. Overall, the work was well-conducted, and I support the publication of the work in Nat. Commun. After the following concerns are addressed.

We are thankful for the positive assessment.

(1) The scope of the allylboronate was not investigated to evaluate the substituent effects of

groups on the alkenyl moiety. More allylboronates are required.

Thank you for this valuable comment. To reflect this suggestion, we conducted extended experiments under optimized condition using two new nucleophilic boronates such as [3-methylbut-2-enylboronic acid pinacol ester (CAS #: 141550-13-2) and allenylboronic acid pinacol ester (CAS #: 865350-17-0)], with diverse ketones [ethyl (2,3,4,5-tetrafluorobenzoyl)acetate (**1a**), acetophenone (**14a**), and 3-pentanone (**10b**)] (see the figure below). According to our experimental results, we could not observe any relevant products under the standard reaction condition, but corresponding hydrazone intermediates were only observed as major by-products. In addition, if we employed a pre-formed hydrazone of **14a**, retro-reaction (hydrolysis) towards **14a** was observed in both boronates. The reason is not clear in the current stage, however, it seems two newly employed nucleophiles are not fit in our postulated transition-state model. We feel further study seems beyond the current submission and can be handled in our future project. To describe the limitation of boronate scope clearly, we added the following sentence in the main text.

“Other boronate nucleophiles such as 3-methylbut-2-enylboronic acid pinacol ester and allenylboronic acid pinacol ester were inactive under the established catalytic condition.”

[a] Reaction condition : ketone (0.1 mmol), benzhydrazide (1.2 equiv.), R-Bpin (1.5 equiv.), DBSA (5 mol%), SQA (5 mol%), PhMe (5 equiv.), H₂O (10 L/mol), 60 °C, 24 h.
 [b] Conversion (conv.) was calculated by ¹H NMR integration. [c] n.d. = not detected. [d] c.m. = complex mixture. [e] p.d. = partial decomposition. [f] hydrazone observation.

(2) It's more desirable if general amines could be used, but it was found aminating sources other than hydrazide are not reactive for this transformation. The authors are suggested to give some comments about why the hydrazide is essential in page 7 about the selection of amine source.

Hydrazide is highly stable and reactive amine in our reaction condition. As we pointed out, we could observe in-situ generated hydrazone during the reaction and even isolable. The other amines rarely afforded the reactive intermediate under the aquacatalytic condition.

Furthermore, as described in the main text, we assumed that hydrazone intermediate might interact with allylboronic acid pinacol ester (**3a**) through Lewis base activation. The control experiments were already described in Fig. 2b. For clarity, we added a sentence as follows.

“Therefore, a water-compatible, reactive benzhydrazide (**2a**) is highly essential for the success of the desired allylation process.”

(3) According to Figure 5B, the alkyl chain on DBSA is not involved in complexation with SQA-III, why this complex was more stable by 2.15 kcal/mol than the corresponding complex with *p*-toluenesulfonate?

We anticipate that a relatively more electron-donating long-chain *p*-alkyl group provides more electron-rich anionic aryl sulfonate than *p*-methyl bounded aryl sulfonate. Therefore, sulfonate of DBSA (conjugate base) in combination with **SQA-III** (a good hydrogen-bonding donor: Brønsted acid) provides a relatively more stable anion-binding complex than that of sulfonate of PTSA. The output of comparative computational calculation supports our hypothesis.

1. shorter non-covalent interaction distance between catalyst and **SQA-III**.
2. bigger average charge (negative value) of sulfonate group.

The comparative data are as follows.

	SQA + p TsOH (sulfonate)	SQA + DBSA (sulfonate)
(N-H) bond distance (Å)	1.042 / 1.044	1.044 / 1.045
interaction distance (Å)	1.695 / 1.709	1.670 / 1.689
average electrostatic charge	-0.189	-0.206
average Mulliken charge	-0.165	-0.168
average natural charge	-0.143	-0.146

(4) The abstract needs revision. Too much background information is given in this section.

We are grateful for this valuable comment. The abstract was revised with key sentences as follows.

“A tetrasubstituted carbon atom connected by three sp^3 or sp^2 -carbons with single nitrogen, i.e., the α -tertiary amine (ATA) functional group, is an essential structure of diverse naturally occurring alkaloids and pharmaceuticals. The synthetic approach toward ATA structures is intricate, therefore, a new and straightforward catalytic method has remained a substantial challenge. Here we show an efficient water-accelerated organocatalytic method to directly access ATA incorporating homoallylic amine structures by exploiting readily accessible general ketones as useful starting material. The unprecedented synergistic action of a hydrophobic Brønsted acid in combination with a squaramide hydrogen-bonding donor under aqueous condition enabled the facile formation of the desired moiety. The developed exceptionally mild but powerful system facilitated a broad substrate scope, and enabled efficient multi-gram scalability.”

REVIEWERS' COMMENTS

Reviewer #2 (Remarks to the Author):

I am satisfied with corrections made by the authors and recommend publication of the revised manuscript in NC.

Reviewer #3 (Remarks to the Author):

I read the revised manuscript twice at different times. The authors have responded to my previous comments properly and I do not have other problems. I support the publication of the work in Nature Communications in its current form.

Reviewer #2 (Remarks to the Author):

I am satisfied with corrections made by the authors and recommend publication of the revised manuscript in NC.

We are grateful for the positive assessment. We highly appreciate the reviewers' insightful and helpful advice on our manuscript.

Reviewer #3 (Remarks to the Author):

I read the revised manuscript twice at different times. The authors have responded to my previous properly and I do not have other problems. I support the publication of the work in Nature Communications in its current form.

We are grateful for the positive assessment. We highly appreciate the reviewers' insightful and helpful advice on our manuscript.